# Impact of Season on Intestinal Bacterial Communities and Pathogenic Diversity in Two Captive Duck Species

**DOI:** 10.3390/ani13243879

**Published:** 2023-12-16

**Authors:** Patthanan Sakda, Xingjia Xiang, Zhongqiao Song, Yuannuo Wu, Lizhi Zhou

**Affiliations:** 1School of Resources and Environmental Engineering, Anhui University, Hefei 230601, China; patthanan423@gmail.com (P.S.); szq202312@163.com (Z.S.); wuyuannuo2022@163.com (Y.W.); 2Anhui Province Key Laboratory of Wetland Ecosystem Protection and Restoration, Anhui University, Hefei 230601, China; 3Anhui Shengjin Lake Wetland Ecology National Long-Term Scientific Research Base, Chizhou 247230, China

**Keywords:** Baikal teal, common teal, intestinal bacteria, potentially pathogenic bacteria, seasonal variations

## Abstract

**Simple Summary:**

The gut bacterial community of two captive ducks was investigated in this study across seasons. It was discovered that the composition of gut bacteria was significantly affected by seasonal variations, with greater diversity in winter. Different seasons yielded different biomarkers, with the majority found in winter when a more intricate bacterial network structure was seen. Ten important pathogenic bacterial species were discovered and appeared to be more abundant in the summer. The presence of significant pathogenic bacteria was demonstrated by this study, raising concerns for captive animals, and shedding light on how gut bacterial composition is affected by seasonal fluctuations.

**Abstract:**

Vertebrates and their gut bacteria interact in complex and mutually beneficial ways. The intestinal microbial composition is influenced by several external influences. In addition to food, the abiotic elements of the environment, such as temperature, humidity, and seasonal fluctuation are also important determinants. Fecal samples were collected from two captive duck species, Baikal teal (*Sibirionetta formosa*) and common teal (*Anas crecca*) across four seasons (summer, autumn, winter, and spring). These ducks were consistently fed the same diet throughout the entire experiment. High throughput sequencing (Illumina Mi-seq) was employed to analyze the V4–V5 region of the 16sRNA gene. The dominant phyla in all seasons were Proteobacteria and Firmicutes. Interestingly, the alpha diversity was higher in winter for both species. The NMDS, PCoA, and ANOSIM analysis showed the distinct clustering of bacterial composition between different seasons, while no significant differences were discovered between duck species within the same season. In addition, LefSe analysis demonstrated specific biomarkers in different seasons, with the highest number revealed in winter. The co-occurrence network analysis also showed that during winter, the network illustrated a more intricate structure with the greatest number of nodes and edges. However, this study identified ten potentially pathogenic bacterial species, which showed significantly enhanced diversity and abundance throughout the summer. Overall, our results revealed that season mainly regulated the intestinal bacterial community composition and pathogenic bacteria of captive ducks under the instant diet. This study provides an important new understanding of the seasonal variations in captive wild ducks’ intestinal bacterial community structure. The information available here may be essential data for preventing and controlling infections caused by pathogenic bacteria in captive waterbirds.

## 1. Introduction

The complex symbiotic relationship between vertebrate hosts and their intestinal microbiota has caused a substantial impact on both organisms’ ecology and evolution [1]. The intestinal microbiota is a diverse community of bacteria, archaea, fungi, and viruses that plays a significant role in maintaining animals’ health [2]. Experimental and comparative research has revealed that the host’s physiology, behavior, immunity, reproduction, and metabolic endocrine system are all balanced by the intestinal microbiome [3,4]. Various factors may affect the intestinal microbial community. According to research on a variety of animals, including Tibetan macaques (*Macaca thibetana*), mesquite lizards (*Sceloporus grammicus*), and Siberian cranes [5,6,7], diet is the main factor influencing the intestinal bacterial composition and function. In addition, seasonal fluctuations and temperature variations can have an impact on the composition and function of the intestinal microbiome [8]. The variations of the intestinal bacterial composition in hosts were found following a steady diet during seasonal change in wild-caught western fence lizards (*Sceloporus occidentalis*), chum salmon (*Oncorhynchus keta*), and homing pigeons (*Columba livia*) [9,10,11]. Particularly in livestock, the intestinal microbial susceptibility to environmental conditions (seasonal variation) is reflected in changed composition and functioning, which is crucial for the host’s general health and welfare status [12]. However, the significant role of host genetics has only been investigated in human research and animal models [13]. In wild animals in the field, microbiome richness and diversity variations have been reported between different hosts. In particular, a significant composition of intestinal and pathogenic bacteria was found in sympatric hooded crane (*Grus monacha*) and greater white-fronted goose (*Anser albifrons*) [14]. 

Baikal teal (*Sibirionetta formosa*) and common teal (*Anas crecca*) are the two dabbling ducks of the Anatidae family. The Baikal teal breeds in eastern Siberia and migrates to China, Japan, and South Korea in winter [15,16,17]. Common teal is a common duck in Asia and Europe, breeding in Eurasia and migrating to wintering grounds in China, Iran, and Pakistan [18,19,20]. Both species have been recognized by BirdLife International and the IUCN on the list of Least Concern (LC). The annual life cycle, including reproduction, migration, and wintering, has enabled these two migratory wild duck species to continuously adapt to new habitats. However, little is known about the intestinal bacterial community in these ducks. 

Therefore, to investigate the effects of both variables (seasonal variations and host species) on the intestinal bacterial community in these animals, they were fed the same food consistently throughout the entire experiment. We conducted a study on the captive Baikal teal and common teal in Shengjin Lake National Natural Reserve, Anhui Province, China, throughout different seasons on an annual basis. Moreover, we investigated potential pathogenic bacterial species. Our study could improve our understanding of the captive duck’s intestinal bacterial community and the impacts of seasonal variations and host species on this community, including the occurrence of pathogenic bacterial communities. This understanding would help us reveal how changing seasons and different hosts interact to shape the intestinal bacterial and pathogenic community and will inform and support conservation practices.

## 2. Materials and Methods

A total of 96 feces samples of captive Baikal teal and common teal (12 samples/duck species/season) were collected from early July 2022 to early April 2023, including four sampling times covering four seasons: 1 July 2022 for summer, 1 October 2022 for autumn, 10 January 2023 for winter, and 1 April 2023 for spring. Fresh feces droppings were collected from individual ducks with a sterile sampling spoon and put in a sterilized plastic Ziplock. Each sample was labeled with the corresponding duck species and the date of collection. All samples were stored at −80 °C until analysis.

These two duck species were bred for a conservation research project at Shengjin Lake, Anhui Province, China during the 2020 year. We provided a consistent commercial duck feed mixed with paddy rice for all seasons, to reduce the influence of dietary fluctuations on the intestinal bacterial community and focus primarily on seasonal variation. 

### 2.1. DNA Extraction, PCR Amplification, and Sequencing 

Following the manufacturer’s instructions, the genomic DNA was extracted using the SPINeasy DNA Kit for Faeces (MP Bio-medicals, Santa Ana, CA, USA). The extracted DNA was quantified using a NanoDrop ND-1000 (Thermo Scientific, Wilmington, DE, USA) and then sent to the Hefei Baisheng Science & Technology Development for library construction, quantitation, pooling, and sequencing. The universal primers 515F (5′-GTGCCAGCMGCC GCGG-3′) [21] and 907R (5′-CCGTCAATTCMTTTRAGTTT-3′) [22] were applied to amplify the V4–V5 region of the 16S rRNA gene. A GeneAmp 9700 thermocycler (Applied Biosystems, Foster City, CA, USA) was employed, to amplify the PCR reaction. The protocol was as follows: denaturation at 95 °C for 3 min, followed by 27 cycles of 95 °C for 30 s, 55 °C for 30 s, and 72 °C for 45 s, and then a final extension at 72 °C for 10 min. We qualified each PCR product using a Qubit 3.0 fluorometer (Thermo Fisher Scientific, Wilmington, DE, USA) before performing next-generation sequencing. Finally, the purified products were sequenced on an Illumina MiSeq PE250 platform (Illumina, San Diego, CA, USA) [23].

### 2.2. Bioinformatics Analysis

The Fast Length Adjustment of SHort reads (FLASH) method was employed to combine the paired-end reads obtained from the sequencing process [24]. Quality filtering of the raw tags was performed under specific filtering conditions according to the QIIME2 to obtain the high-quality clean taq, and amplicon sequencing chimeras were removed by VSEARCH [25]. The extracted sequences were clustered by 100% similarity to the amplicon sequence variants (ASVs). The taxonomy was classified based on the SILVA 132 database [26]. A randomly selected subset of 12,800 sequences per sample was used to compare the bacterial community between seasons and duck species.

### 2.3. Potentially Pathogenic Bacterial Identification

To identify potentially pathogenic bacteria in the captive Baikal teal and common teal, a manual search was conducted utilizing the identified species result combined with keywords, such as “pathogenic bacteria” and “bacterial pathogens”, in the Web of Science, ScienceDirect, and PubMed platforms. Based on references, bacterial species that have been proven to be pathogens in humans and/or animals were chosen for further analysis. This investigation revealed a total of 10 species of potentially pathogenic bacteria, as shown in Appendix A.

### 2.4. Statistical Analysis

Venn diagrams were constructed based on the ASVs showing the number of shared and different ASVs between groups. Alpha diversity was applied to analyze the species diversity complexity of a sample through two indices (observed species and PD whole tree). For the beta diversity, we conducted Non-metric Multidimensional Scaling (NMDS) based on Bray–Curtis dissimilarity matrices and Principal coordinate analysis (PCoA) at the ASV level based on unweighted UniFrac distances, to visualize the bacterial community between different seasons and duck species. An Analysis of similarities (ANOSIM) was used to identify the differences in bacterial community structure among all sample groups. The LEfSe (Linear discriminant analysis Effect Size) algorithm was performed to identify the specific taxa that exhibited the most significant differences across sample groups. Bar plots and a cladogram were used to graphically illustrate differentially abundant taxa that surpassed an LDA log score of 2.0 and where the *p*-value was <0.05. The LEfSe analysis was restructured using the Huttenhower Lab Galaxy Server [27]. The relative abundance was calculated by the equation: relative abundance (%) = [(number of sequences of each ASV ÷ total sequences per sample) × 100]. The ggplot2 packages in the R program were employed to generate plots of relative abundance and differences in alpha and beta diversity.

### 2.5. Co-Occurrence Network Analysis and Keystone Taxa

The microbial co-occurrence networks based on ASVs were constructed to obtain insight into potential bacterial associations within the seasonal variation. All pairs of ASVs were compared using the Spearman correlation approach for each network, and the *p*-values were altered using the Benjamini and Hochberg (BH) methods for false discovery [28]. The topological role of each ASV, within connectivity (Zi) and among module connectivity (Pi), was computed. Each ASV’s topological role was determined according to the suggested Zi and Pi degree thresholds [29]. Based on the Zi and Pi values, the network nodes were divided into four groups: peripherals (Zi < 2.5 and Pi < 0.62), connectors (Pi ≥ 0.62 and Zi < 2.5), network hubs (Zi ≥ 2.5 and Pi ≥ 0.62), and module hubs (Zi ≥ 2.5 and Pi < 0.62). The nodes in the connectors, network hubs, and module hubs were determined to be the keystone species in the microbial community [30]. Overall, the co-occurrence networks were assessed using the R package igraph 1.2.6. The visualizing networks and calculating topological characteristics were conducted on Gephi software (version 0.10.1) [31], where nodes were colored by phylum, and the size of the nodes was scaled to their group abundance.

## 3. Results

### 3.1. Intestinal Bacterial Alpha Diversity

We obtained 3,403,674 high-quality reads for downstream analyses, ranging from 12,850 to 63,959 sequences per sample. The processed data revealed 26 phyla, 40 classes, 95 orders, 203 families, 453 genera, and 3,731 bacterial ASVs. Across all samples, 70 ASVs (1.88%) were found in four seasons in two duck species. The sample of the summer Baikal teal (SMB), summer common teal (SMC), autumn Baikal teal (ATB), autumn common teal (ATC), winter Baikal teal (WTB), winter common teal (WTC), spring Baikal teal (SPB), and spring common teal (SPC) had 405 (14.73%), 585 (17.57%), 217 (7.89%), 102 (3.06%), 553 (20.11%), 921 (27.67%), 307 (11.17%), and 238 (7.15%) unique bacterial ASVs, respectively. For each duck species, the Baikal teal shared 136 ASVs (4.95%) across four seasons, while the common teal shared 103 ASVs (3.09%). Meanwhile, both duck species showed the highest number of unique ASVs in winter, followed by summer, spring, and autumn, respectively (Figure 1A,B).

In order to assess the alpha diversity of intestinal bacteria in Baikal teal and common teal across four seasons, the Observed species and PD whole tree were utilized. Significant differences (*p* < 0.05) in the diversity indices across seasons and duck species were tested using the One-Way ANOVA with Duncan test. The common teal showed a significantly highest observed species in winter, lowest in autumn, and no variation between summer and spring. The Baikal teal results indicated no significant difference across four seasons. In addition, autumn and winter showed significance within duck species, with Baikal teal higher in autumn and lower in winter (Figure 2A). The PD whole tree, on the other hand, varied across different seasons and duck species. The Baikal teal showed a significantly higher level in winter and spring and no difference between summer and autumn. While common teal showed a significant decrease in autumn and a significant increase in winter (Figure 2B).

### 3.2. Intestinal Bacterial Community Structure

The dominant bacterial phyla in captive Baikal teal and common teal across four seasons were Proteobacteria (41.24%), Firmicutes (34.44%), Bacteroidetes (18.34%), Actinobacteria (2.94%), Cyanobacteria (1.48%), Verrucomicrobia (0.61%), Fusobacteria (0.55%), and Epsilonbacteraeota (0.11%) (Appendix A). During the summer and autumn seasons, Proteobacteria showed up in prominent abundance in both duck species. Contrarily, the spring season revealed higher Firmicutes in both ducks (Figure 3A,B). The abundance of Proteobacteria in Baikal teal was significantly higher during the winter season, with no difference in other seasons. Conversely, the common teal exhibited a significant decrease in Proteobacteria during the spring season, with no discernible variations found throughout the other seasons. Firmicutes appeared significantly in both duck species in spring, and there were variances between the two duck species that were greater in common teal. In Baikal teal, Bacteroidetes exhibited a decline in the winter but no significant variation in other seasons. Whereas common teal showed a considerable rise in autumn and no difference in other seasons. In both duck species, Actinobacteria were much lower in the spring, similar in other seasons for Baikal teal, and significantly higher in the summer for common teal (Appendix A).

Linear discriminant analysis (LDA) effect size (LEfSe) analysis was performed to further identify specific taxa changes in the intestinal bacterial compositions between two duck species across four seasons using phylum-to-order level data. A total of 25 bacterial taxa exhibited LDA scores greater than two during all seasons. The result showed that winter exhibited a higher significant abundance of bacterial taxa compared to other seasons, especially in common teal, which had an abundance of three phyla (Cyanobacteria, Verrucomicrobia, and Tenericutes), five classes (i.e., Verrucomicrobiae, Coriobacteriia, Deltaproteobacteria), and six orders (i.e., Nostocales, Verrucomicrobiales, Coriobacteriales). On the other hand, Baikal teal revealed a significant abundance of only one phylum (Proteobacteria). While one phylum (Fusobacteria), one class (Fusobacterii), and two orders (Fusobacteriales and Aeromonadales) were significantly higher throughout the spring season in Baikal teal, one phylum (Acidobacteria), one class (Blastocatellia), and three orders (Chitinophagales, Leptolyngbyales, and Blastocatellales) were more apparent in Baikal teal during the autumn season. However, only one phylum (Enterobacteriales) was shown to be more abundant in Baikal teal throughout the summer. Notably, no significantly enhanced bacterial taxa were seen in common teal throughout the spring, autumn, or summer seasons (Figure 4A,B).

At the genus level, a total of 453 genera were identified across the samples and 19 genera were predominant genera (relative abundance > 1.0) (Appendix A). The predominant genus for the Baikal teal varied according to the seasons. *Escherichia-Shigella* and *Sphingobacterium* were the dominant genera in summer and autumn, whereas *Psychrobacter* and *Chryseobacterium* were dominant in winter and spring. Common teal, on the other hand, showed different dominating genera for each season. In particular, *Escherichia-Shigella* and *Lactobacillus* dominated throughout summer, *Chryseobacterium* and *Sphingobacterium* were dominant during autumn, *Psychrobacter* and *Ralstonia* dominated during winter, and *Enterococcus* and *Clostridium sensu stricto 1* dominated during spring (Appendix A).

To further assess the overall composition of the intestinal bacterial communities during four seasons in the captive Baikal teal and common teal, the beta diversity was analyzed using NMDS and PCoA analysis plots, based on the ASV level, to visualize the differences between different seasons and duck species. NMDS and PCoA showed distinct variations in intestinal bacterial composition among four seasons. However, there were no significant differences between the duck species (Appendix A). In Baikal teal, both NMDS and PCoA revealed more similarities between the winter and spring seasons (Figure 5A,B). While in common teal, NMDS analysis exhibited more similarities between the winter and spring seasons, whereas PCoA showed more similarities between the autumn and summer seasons (Figure 5C,D).

### 3.3. Co-Occurrence Pattern and Keystone Taxa of Intestinal Microbial Communities

Network analysis was utilized to investigate the microbial co-occurrence pattern (Figure 6). Overall, the intestinal microbiota of Baikal teal (1061 nodes; 70,271 edges) and common teal (1252 nodes; 72,826 edges) had the most complex network during winter, followed by spring (SPB: 952 nodes, 41,618 edges; SPC: 961 nodes, 58,906 edges), summer (SMB: 829 nodes, 31,944 edges; SMC: 945 nodes, 40,447 edges), and autumn (ATB: 925 nodes, 41,290 edges; ATC: 525 nodes, 15,842 edges). Several other important network topological properties, such as density, diameter, modularity, average degree, and the number of communities, also showed differences in the network structures of four seasons in both duck species (Appendix A).

The keystone taxa varied among different seasons and duck species. During summer, the prominent keystone taxa in Baikal teal mostly belonged to the genera *Pseudomonas* and *Sphingobacterium*, while in common teal was *Psychrobacillus*. In autumn, *Sphingobacterium* was primarily in Baikal teal, while *Chryseobacterium* and *Oerskovia* were in common teal. In winter, *Erysipelotrichaceae* and *Pseudomonas* were prominent in Baikal teal, and *Enterococcus* and *Lactobacillus* were in common teal. Finally, in spring, keystone taxa in the genus *Peptococcus* were represented in both duck species (Appendix A, Supplementary Material).

### 3.4. Potential Pathogenic Bacterial Community

We identified 10 potential bacteria during our examination of the intestinal bacteria in Baikal teal and common teal across four seasons. The dominant species observed was *Empedobacter falsenii,* followed by *Stenotrophomonas koreensis*, *Empedobacter brevis*, and *Arsenicicoccus dermatophilus*. The total number of all potential pathogenic bacteria sequences across the four seasons in the two duck species was 15,753 ASVs or approximately 0.46% of the total number of bacteria. Each sample contained a different number of pathogenic bacteria sequences, ranging from 0 to 3521 sequences per sample (Appendix A). One-way ANOVA and Duncan tests were employed to test the significant difference (*p* < 0.05) in the relative abundance and diversity of potentially pathogenic bacteria between different groups. Results indicated that potentially pathogenic bacteria were substantially more abundant in the summer (SMB, SMC), especially in Baikal teal. However, no difference in abundance was observed in the other three seasons (Figure 7A). Regarding pathogen diversity, we found that both Baikal teal and common teal had a significantly higher diversity of pathogens in summer. The next-highest diversity was shown by Baikal teal in autumn, with no significant difference in diversity seen in winter or spring (Figure 7B). 

The relative abundance of each pathogenic species across different groups is shown in Appendix A. Specifically, *Empedobacter falsenii* and *Flavobacterium ceti* showed significantly higher levels in the SMB group, whereas the *Arsenicicoccus dermatophilus*, *Paracoccus yeei*, and *Stenotrophomonas koreensis* levels were significantly higher in the SMC group. Additionally, *Campylobacter canadensis* was revealed to be significantly higher in the SPC group.

## 4. Discussion

The intestinal microbiota is a complex ecosystem composed of billions of bacteria that are constantly altered by a variety of factors, including dietary preferences, seasonality, lifestyle, stress, the use of antibiotics, and illnesses [32]. Our study revealed a strongly significant relationship between season and the taxonomic composition of the intestinal bacterial communities of captive Baikal teal and common teal reared in the same conditions. The bacterial composition, predominant bacteria, alpha and beta diversity, biomarkers, key taxa, and potential pathogens exhibited variation and specificity across different seasons. These findings provide the initial evidence highlighting the significant influence of seasonal variation on the intestinal bacterial community and the occurrence of potentially pathogenic bacteria in the captive Baikal teal and common teal. 

### 4.1. The Intestinal Bacterial Composition

In general, the intestinal bacterial composition in the captive Baikal teal and common teal exhibited similarities to previous findings in the microbiome of both wild birds and livestock. The dominant phyla were Firmicutes and Proteobacteria, with a lower abundance of Bacteriodetes and Actinobacteria [33]. These phyla have been determined to be crucial for the immunity, metabolism, and nutritional absorption of wild birds. For example, Firmicutes and Bacteroidetes contribute to hosts by digesting fatty acids and degrading complex carbohydrate polysaccharides [34,35]. The main diet we provided was commercial feed mixed with paddy rice, which is a great supply of complex carbohydrates [36], which is connected to the high abundance of Firmicutes. Moreover, the bacterial community is dominated by Proteobacteria, which are a diverse group of species with varying metabolic capabilities to break down organic substances for energy [37]. Actinobacteria are essential for the carbon cycle and the cycling of soil organic matter [38], but there is no evidence to support Actinobacteria’s role in domestic or wild birds [33]. However, compared with other wintering waterbird species in the same location, such as hooded crane, greater white-fronted goose, bean goose, and lesser-fronted goose, the Firmicutes/Proteobacteria ratio in the present investigation was different [14,35,39,40,41,42,43]. These migratory birds exhibited a significant peak level of Firmicutes, attributed to their diet primarily consisting of rice fields and meadows [44]. On the other hand, our study showed a higher average abundance of Proteobacteria (Appendix A). Furthermore, this investigation identified *Escherichia-Shigella* (16.28%), a genus member of the family *Enterobacteriaceae*, as the significantly prominent genus. Compared to other results, *Escherichia-Shigella* showed a marked variation, with a lower abundance than Red Knot (*Calidris canutus*) and Ruddy Turnstone (*Arenaria interpres*) birds [45], while showed a higher abundance than that observed in Shaoxing ducks bred with a commercial diet within the local farm of Wenzhou City, China [46]. However, several variables can alter the composition of gut bacteria in captive animals, including dietary changes, medical interventions, and increased isolation from humans and other animals [47].

### 4.2. Seasonal Variation Effect

Seasonal variations in the gut microbiota have been reported in a variety of vertebrate lineages, including birds, reptiles, and mammals. Wild bird studies, such as those of hooded cranes, passerine birds, *Catharus thrushes*, and Sichuan partridges, have demonstrated that the seasonal modulation of the gut microbiota often corresponds to seasonal changes in diet [42,48,49,50]. While research on homing pigeons given the same diet in the Netherlands suggests that temperature played a role in the seasonal variations in the composition of the gut bacterial community [11], this study presents the first evidence of seasonal effects on the gut bacterial community in captive wintering ducks in China.

During summer, there was a significant decrease in Firmicutes, aligning with recent experimental findings in reptiles and mammals, such as lizards (*Zootoca vivipara*), house mice (*Mus musculus domesticus*), and cattle (*Bos taurus*). Experimental evidence suggests that rising temperatures contribute to a decrease in the abundance of Firmicutes [51,52,53]. Similarly, laying hens under conditions of high heat stress also showed a significant decrease in the relative abundance of Firmicutes [54]. However, LEfSe results showed that the order Enterobacteriales, belonging to the phylum Proteobacteria, was identified as the biomarker of the summer season, consistent with the previous finding that Enterobacteriales predominated in the summer [55]. 

In winter and spring the ducks exhibited a significant decrease of Bacteroidetes, consistent with previous reports in reared worms (*Caenorhabditis elegans*) and house mice showing the decreasing abundance of Bacteroidetes in cold weather [8,56]. However, the bioindicators in winter were more numerous than in other seasons, which also varied between different hosts. The significant abundance of Phylum Proteobacteria in Baikal teal in winter may be connected to the previously described enriched Proteobacteria in frogs in low temperatures [57]. Additionally, previous studies in mice in controlled environments showed that exposure to cold temperature dramatically increased the phylum Proteobacteria, and this shift aided the host in enduring periods of high energy demand that are typical of cold seasons [52]. Phylum Cyanobacteria and order Nostocales were the most enriched in common teal, which were also shown as biomarkers in hooded cranes in the winter season [41]. Temperature and light intensity are key factors in the regulation of the growth of Nostocales [58].

Furthermore, our study revealed the difference in alpha diversity of the intestinal bacterial communities during different seasons in both duck species, which was significantly higher in the winter season, and different between duck species. Consistent with previous research on hooded cranes, alpha diversity was significantly higher in winter than in autumn and spring [42]. It is hypothesized that the microbiome shows flexibility when faced with environmental changes, which is consistent with other research that highlights the significant influence of environmental conditions and cohabitation on intestinal microbiota communities [59]. These changes include biotic factors, like the social environment, community makeup, and the quantity and distribution of vectors, as well as abiotic factors like temperature, moisture, pH, sun radiation, and seasonality [60].

### 4.3. Co-Occurrence Network and Key Taxa

Co-occurrence network analysis, as measured by correlations between abundances of microbial taxa, has proven helpful in comprehending intricate microbial interaction patterns [61]. Microbial network analysis may show how certain species coexist in specific niches and interact with environmental factors [62]. Additionally, it can identify the ‘keystone’ taxa that have the greatest impact on communities, which may play a role in contributing to ecosystem stability [63,64]. This is the first study exploring the intestinal bacterial co-occurrence network and key taxa in captive ducks across different seasons. Our results found that the number of nodes, edges, modularity, average degree, average clustering coefficient, and average path length were different between different seasons and duck species. Intriguingly, the percentage of Proteobacteria nodes in the network decreased with decreasing temperature, while Firmicutes increased with decreasing temperature. The microbial community regularly shifts in terms of stability and structure according to seasonal environmental variables [65,66], which could result in different networks and keystone taxa in our study throughout four seasons. As a result, the winter season exhibited the most complexity, characterized by the highest number of nodes and edges, the highest average degree, and the lowest average path length. Notably, the key taxa *Erysipelotrichaceae* displayed consistency with previous research on the intestinal microbiome of broiler chickens, which revealed that it was more prevalent during the winter [67]. Similar patterns were observed in cockatiels (*Nymphicus hollandicus*), which were dominated by the family *Erysipelotrichaceae* [68]. Additionally, the key taxon, *Enterococcus* species identified in this research, was found to be consistent with the study of farm swan goose (*Anser cygnoides*) at Shengjin Lake. This previous investigation also recognized *Enterococcus* as a key microbe [69]. However, determining the relative contribution of environmental conditions to keystone taxa, and keystone taxa to community stability, is challenging [70]. Thus, in this study, we explored the effect of the variation of seasonal factors on bacterial community stability in captive ducks. 

### 4.4. Pathogenic Bacteria

The present study also detected pathogenic bacterial sequences in Baikal teal and common teal, accounting for 0.46% of the total bacterial sequences read. A total of 10 bacterial species were identified as pathogenic bacteria linked to human and animal diseases, including pododermatitis, diarrhea, malaise, nausea, vomiting, hyperacute, and hemorrhagic septicemia (Appendix A). Waterbirds, especially ducks, have been identified to be carriers of several diseases as they favor freshwater environments, where their defecation establishes them as the primary spreaders of pathogens [71]. 

*Empedobacter falsenii* was the most prevalent pathogen species in the present study, as evidenced by several case reports in a variety of materials, including blood cultures, stool [72,73], hospital carpets [74], as well as rodent skins [75]. This bacterial species might be a source of resistance genes in hospitalized patients who are immunocompromised or receiving medical treatment [68]. Another dominant pathogen, *Sphingomonas koreensis*, is commonly found in aquatic environments like mineral water [76]. However, it has also been reported as the causative agent of meningitis and polymicrobial peritonitis in humans [77,78]. *Dietzia maris* is another significant pathogen, which is regarded as an opportunistic infection depending on a variety of factors including the presence of foreign material (prosthetic hip) [79], the patient’s immunological status, or epidemiological risk factors, such as animal interaction and work-related exposure to animal farms [80]. However, *Dietzia maris* was significantly abundant in outdoor laying hens, which related to the contact with soil [81]. Although there is no confirming evidence that these species cause illnesses in animals, further studies should be executed to examine the possible effects of these pathogens on wild animals or poultry’s health. However, *Campylobacter* species are one of the main clinical pathogens and important contributors to bacterial gastrointestinal illnesses and present an economic and public health concern worldwide [82]. *Campylobacter canadensis* was first discovered in captive whooping cranes at the Calgary Zoo in Canada [83]. Although various wild birds are known as the sources of *Campylobacter* [84], poultry is accepted to act as the main reservoir of human infections, as previous research showed that the abundance of *Campylobacter* was greater in commercially raised turkeys than in wild turkeys [85]. Therefore, the infection of *Campylobacter* is principally associated with the consumption of products from the poultry chain, and because of this, it is crucial to keep pathogens under control on the farm [86]. 

However, potential pathogens’ relative abundance and diversity were significant between seasons. The samples in summer showed a significantly higher abundance and diversity than in other seasons, which is consistent with a previous study that exhibited that high temperature could facilitate the growth and pathogenicity of some pathogenic bacteria [87,88,89]. While the diversity of the bacterial community was substantially larger in winter, this was consistent with prior findings that suggested that the bacterial community and potential pathogens were uncorrelated favorably [14,43].

Nevertheless, our research suggests a potential relationship between host species and pathogenic bacteria. The evidence is the difference in pathogenic bacteria observed between the two duck species, Baikal teal exhibited higher average pathogenic ASVs and the percentage of pathogenic sequences than common teal. That supports the hypothesis that pathogen reservoirs may be caused by differences in the physiological phenotype of the host [90], and that host heterogeneity has a substantial impact on host susceptibility to infectious illness [91]. The current findings in captive ducks exhibited a reduced presence of the pathogen bacteria to previous research on wild waterbirds and also fluctuations in the dominant species [41]. The main reason for higher infection rates in wild animals is the environmental contamination sharing, such as habitat and food, especially water resources, with other animals that support the transmission from one species to another [92]. In summary, the results suggest that more attention should be paid to the potential pathogens of poultry and the wild waterbirds that migrate to this area, to prevent disease transmission in conspecifics and other mixed species.

## 5. Conclusions

Our research provided the first understanding of the intestinal bacterial composition and potential pathogenic bacteria across seasonal variations in captive Baikal teal and common teal. Our results showed that the four dominating phyla were Proteobacteria, Firmicutes, Bacteroidetes, and Actinobacteria in all seasons, and also revealed the fluctuation of each phyla in different seasons. However, there was an apparent distinction between Baikal teal and common teal in the abundance of Proteobacteria and Firmicutes, with Baikal teal having more abundance of Proteobacteria and fewer Firmicutes than common teal. In the winter season, the alpha diversity of the bacterial community experienced a notable increase, particularly in common teal. This was accompanied by the presence of the highest number of significantly different taxa through LefSe analysis. Additionally, the co-occurrence networks exhibited the highest level of complexity, as indicated by a greater number of nodes and edges during this season. Conversely, during the summer season, higher abundance and diversity were noted for ten pathogenic bacterial species, especially in Baikal teal. The most prevalent species among them was *Empedobacter falsenii*. These results provide useful insights into the seasonal variation that plays a vital role in shaping the intestinal bacterial community and is crucial in preventing severe infections caused by pathogenic bacteria in captive waterbirds. However, our study had a limited observation about other abiotic factors, and we suggest gathering of more information on abiotic factors like temperature, day length, and humidity index to strengthen the validity of future studies.

## Figures and Tables

**Figure 1 animals-13-03879-f001:**
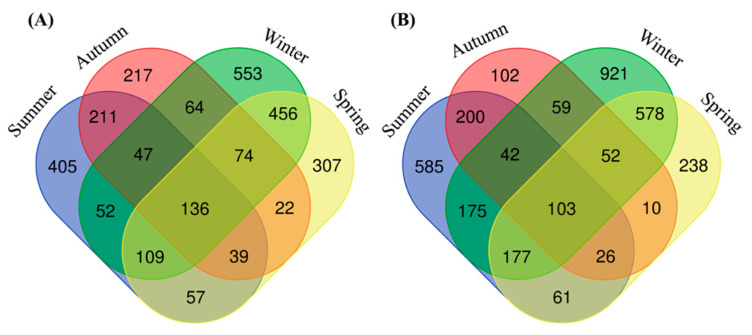
Venn diagrams show unique and shared bacterial ASVs across four seasons in (**A**) Baikal teal and (**B**) common teal.

**Figure 2 animals-13-03879-f002:**
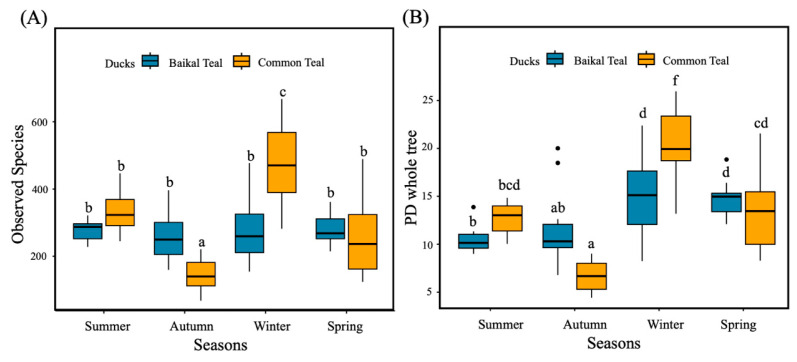
Alpha diversity of intestinal bacterial communities in captive Baikal teal and common teal during four seasons. (**A**) Boxplot for comparison of species richness (Observed species) and (**B**) Boxplot for comparison of phylogenetic diversity (PD whole tree). Data were analyzed using One-way ANOVA with Duncan test and difference letters represent the significant difference (*p* < 0.05).

**Figure 3 animals-13-03879-f003:**
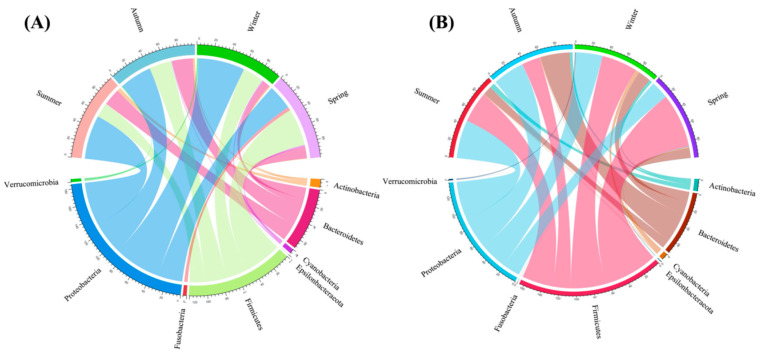
The chord diagram illustrates the proportionate distributions of the dominant bacterial phyla in two captive ducks during four seasons. (**A**) Dominant phyla in Baikal teal and (**B**) dominant phyla in common teal.

**Figure 4 animals-13-03879-f004:**
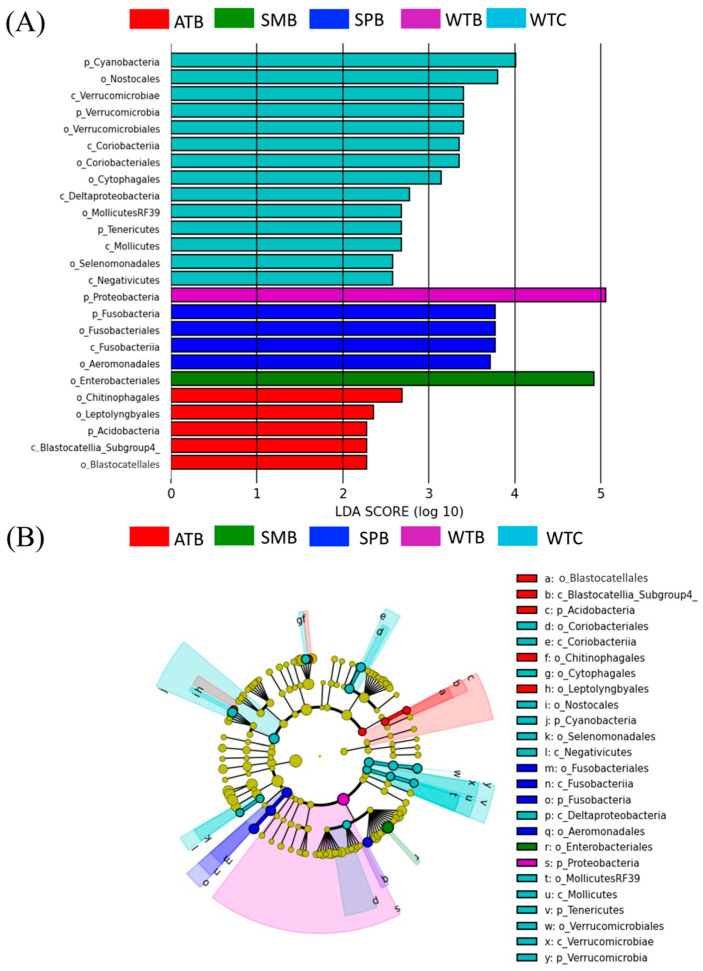
Linear discriminant analysis effect size (LEfSe) analysis of bacterial abundance differences among groups. The Kruskal-Wallis test (0.05) and LDA threshold score of 2.0 obtained the differences among classes. (**A**) The bar plots represent the significantly differential taxa between groups, based on effect size (LDA score). (**B**) The cladogram shows the difference in enriched taxa between groups. ATB (autumn Baikal teal), SMB (summer Baikal teal), SPB (spring Baikal teal), WTB (winter Baikal teal), and WTC (winter common teal). The letters in front of OTUs represent taxonomic levels (p = phylum, c = class, and o = order).

**Figure 5 animals-13-03879-f005:**
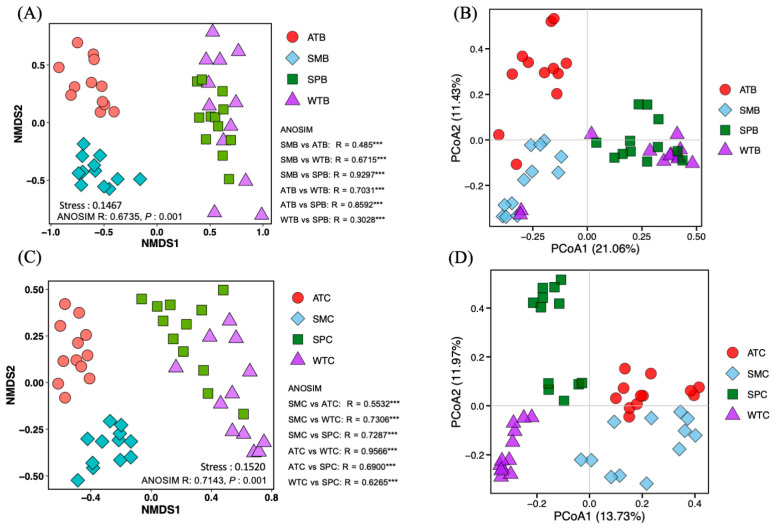
The community structure of bacterial communities during four seasons in captive dabbling ducks was assessed by beta diversity patterns using NMDS, PCoA analysis, and ANOSIM R-statistics. The significant difference was denoted with asterisks (*** *p* < 0.001). (**A**) NMDS in Baikal teal, (**B**) PCoA in Baikal teal, (**C**) NMDS in common teal, and (**D**) PCoA in common teal. SMB (summer Baikal teal), SMC (summer common teal), ATB (autumn Baikal teal), ATC (autumn common teal), WTB (winter Baikal teal), WTC (winter common teal), SPB (spring Baikal teal), and SPC (spring common teal).

**Figure 6 animals-13-03879-f006:**
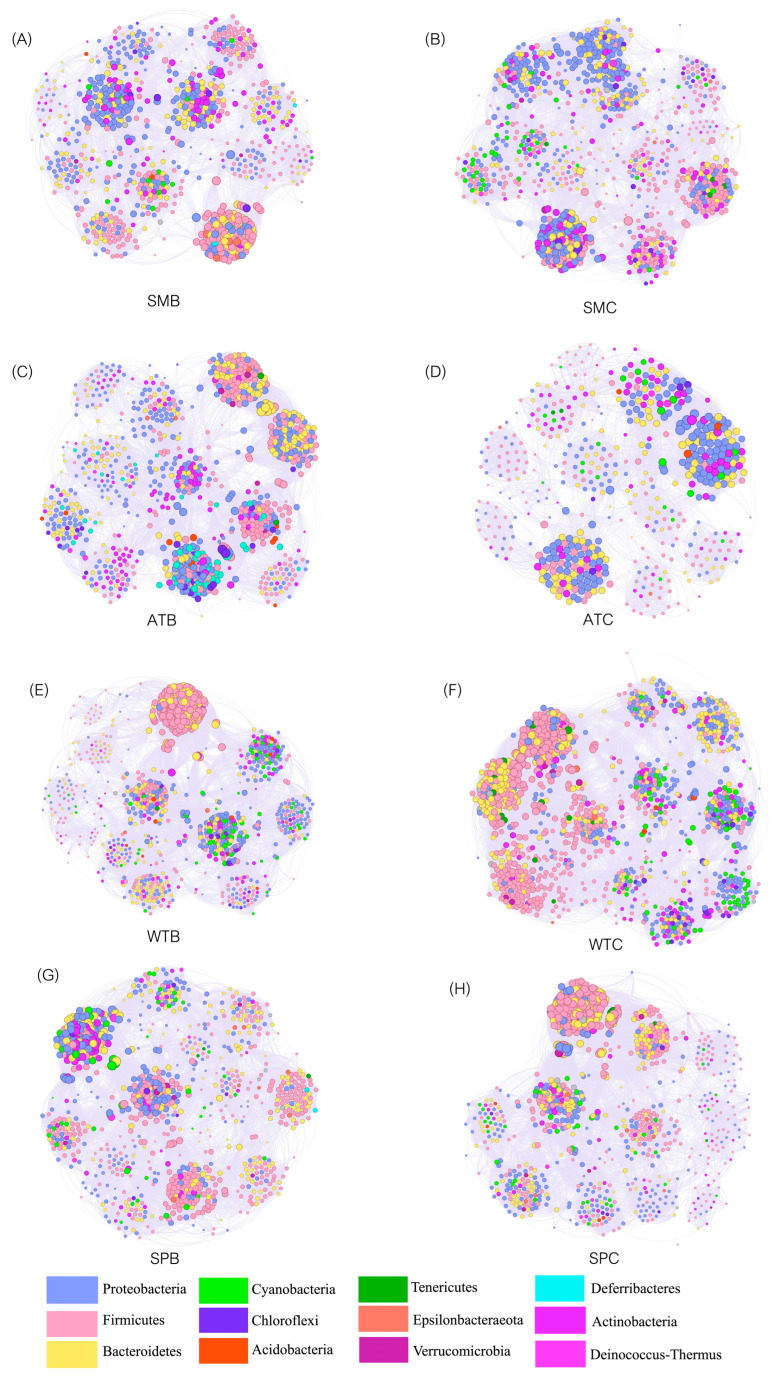
Co-occurrence patterns of bacterial communities in different groups. (**A**) SMB (summer Baikal teal); (**B**) SMC (summer common teal); (**C**) ATB (autumn Baikal teal); (**D**) ATC (autumn common teal); (**E**) WTB (winter Baikal teal); (**F**) WTC (winter common teal); (**G**) SPB (spring Baikal teal); and (**H**) SPC (spring common teal). The size of each node is proportional to the number of degrees.

**Figure 7 animals-13-03879-f007:**
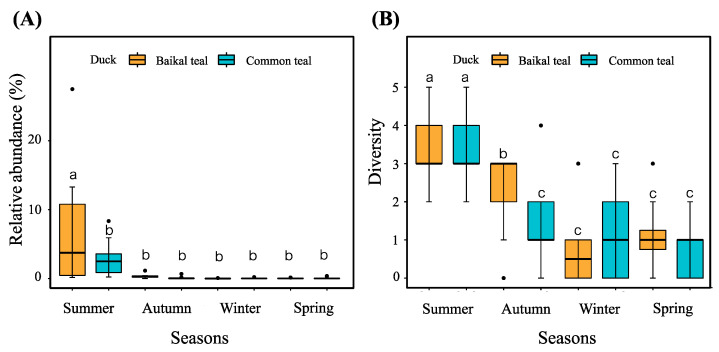
The differences in (**A**) relative abundance and (**B**) diversity of potentially pathogenic bacteria in captive Baikal teal and common teal during four seasons. Data were analyzed using One-way ANOVA with Duncan test and difference letters represent the significant difference (*p* < 0.05).

## Data Availability

The raw data have been submitted to the NCBI Sequence Read Archive (BioProject identifier (ID): PRJNA1025959, PRJNA1025961).

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
