# Peer review of "Impact of Season on Intestinal Bacterial Communities and Pathogenic Diversity in Two Captive Duck Species"

_animals, 2023, doi:10.3390/ani13243879_

Round 1

Reviewer 1 Report

Comments and Suggestions for Authors

L. 85: Set the exact number of the two subgroups

L. 88-90: Explain the sampling method.

L. 92-94: Please, give the exact composition of the diet

L 101-102: References?

L. 109 Bioinformatics analysis

Which platform did they use? Were sequences trimmed or excluded as expected errors? About Operational taxonomic units (OTUs), in which similarities were clustered? Did you remove any OTUs with low relative abundance?  Accession numbers for sequencing analyses?.

The figures have a low resolution in the version available.

The section Conclusions need enrichment and same data addition.

Author Response

Response to reviewer 1# comments

Dear editor, associate editor, and reviewer

We hope this letter finds you well. I am writing to submit the revised version of our manuscript titled “Seasonal Fluctuations Impact on Intestinal Bacterial Communities and Pathogenic Diversity in Two Captive Dabbling Ducks” which was initially submitted to “Animals” under the manuscript ID animals-2722802. We sincerely thank the reviewers for the insightful feedback and valuable suggestions provided during the review process.

We have carefully addressed each reviewer's comments and suggestions and all the changes are highlighted with yellow color in this revised version. In the following, we have attached a detailed response to the reviewers' comments, outlining how each comment was addressed, and would like to highlight the major changes we have made as follows:

Reviewer 1#

  1. Comment: 85: Set the exact number of the two subgroups

Response: We are very grateful for your suggestion. The exact number of the subgroups has been added. Please refer to the updated version for your review, Lines 87-88.

  1. Comment: 88-90: Explain the sampling method.

Response: Thank you for your thoughtful comment. The sampling method has been described according to your comment. Please refer to the updated version for your review, Lines 90-94.

  1. Comment: 92-94: Please, give the exact composition of the diet

Response: We appreciate your suggestion. The exact composition of the diet is paddy rice, and we already added it to the sentence. Please refer to the updated version for your review, Lines 96-97.

  1. Comment: L 101-102: References?

Response: Thanks for the reviewer's kind suggestions. The references for using primers have been added in the “Materials and Methods” and “ References” sections following your suggestion. Please refer to Line 105 and the references section [21-22] in the updated version for your review.

  1. Comment: 109 Bioinformatics analysis

Which platform did they use? Were sequences trimmed or excluded as expected errors? About Operational taxonomic units (OTUs), in which similarities were clustered? Did you remove any OTUs with low relative abundance?  Accession numbers for sequencing analyses?.

Response: Thank you for your thoughtful comment. We did not utilize Operational Taxonomic Units (OTUs); instead, our approach was based on "Amplicon Sequence Variants (ASVs)" at 100% sequence similarity. The raw sequences underwent processing through the QIIME2 pipeline and the sequence errors were removed from amplicon reads with the denoising step. We removed ASVs with relative abundances less than 0.01% of the total number before processing the co-occurrence network to reduce rare ASVs in the data set. The accession numbers for all sequences have been submitted to the NCBI Sequence Read Archive (BioProject identifier (ID): PRJNA1025959, PRJNA1025961).

  1. Comment: The figures have a low resolution in the version available.

Response: Thank you for your kind suggestions. The resolution of Figures 5, 6, and 7 has been enhanced. Please refer to the updated version for your review.

  1. Comment: The section Conclusions need enrichment and same data addition.

Response: Thanks to the reviewer for pointing out the improvement. The section conclusion has added more data and revised it according to your suggestion. Please refer to the updated version for your review, Lines 488-494.

We try our best to improve the manuscript and make some changes in the manuscript. And we have marked all the changes in yellow highlighted in the revised manuscript. We appreciate the Editors/Reviewers’ warm works earnestly and hope that the correction will meet with approval.

Once again, thank you very much for your comments and suggestions. I believe that these revisions have significantly enhanced the quality and contribution of our manuscript. We hope that these changes align well with the expectations of Animals and its readers. Please feel free to contact us if you need any further information or clarification. We are ready to provide any additional materials that may assist in the review process.

Thank you for your time and consideration.

Patthanan Sakda 1,2, Xingjia Xiang 1,2,3*, Zhongqiao Song 1, Yuannuo Wu1,2, and Lizhi Zhou 1,2,3*

1 School of Resources and Environmental Engineering, Anhui University, Hefei 230601, China

2Anhui Province Key Laboratory of Wetland Ecosystem Protection and Restoration, Anhui University, Hefei 230601, China

3Anhui Shengjin Lake Wetland Ecology National Long-Term Scientific Research Base, Chizhou 247230, China

* Correspondence: [email protected] (X.X.); [email protected] (L.Z.)

Reviewer 2 Report

Comments and Suggestions for Authors

Dear authors,

in your text you presented a study related to the bacterial microbiota  of two species of ducks kept in captivity, during the four seasons.  The advantage of this research is certainly the presentation of the microbiota through all seasons, and in two species of birds, kept in the same conditions, using modern molecular methods. The disadvantage is certainly incomplete data about the birds themselves - it cannot be estimated from the text itself whether they are two birds or several individuals of two different species of ducks, which makes the work somewhat incomprehensible.

Specific comments:

Wording "seasonal fluctuations" - please be more specific- fluctuation of what? temperature, humidity...or use only word "season" in title

It is necessary to correctly write the names of bacteria, genera, families, etc. throughout the text; write italics where necessary.

Determination of bacteria as "pathogenic" -  why bacteria of the genus Escherichia-Shigella were not classified as pathogenic? 

Title - please rephrase (possibly as "impact of season on..." and also please be more specific "two captive dabbling ducks" or two captive duck species"?

L 31-32 please rephrase this sentence (it is not clear "winter had sophisticated network structure"?)

L 37-38 please rephrase ("severe infection from the harmful diseases"?)

Chapter 2. Material and Methods - there is a clear need for more information regarding the birds and sampling (how many birds were used in this experiment (in L 91 it is stated "these two ducks")? how were they kept-inside, outside? what was their diet (only rice; L 320, 334?)?

L 89 what does it means that the samples were "suddenly put"?

L 106-108 please check, verb missing, reference?

L 115 classifier?

L 133-134 please check - "that most differentially across"?

L 162-165 please first write in full and than abbreviation in brackets

L 198-200 please check - it is not clear "common teals exhibited a significant decrease"?

L 202 please check (it is only written "in Baikal teal"?)

L 236 - please check (is the number of total of genera missing in this sentence?)

L 265-267 please check this sentence

L 357 -please change "variation of seasonal variation"

In Discussion there are multiple sentences that are not clear enough and should be rephrased (L 333-369, L 373-373, L 379-381, L 410-412, L 431-433, L 438-441, L 462-465)
L 451 please change "is much higher level"

Comments on the Quality of English Language

Minor changes needed

Author Response

Response to reviewer 2# comments

Dear editor, associate editor, and reviewer

We hope this letter finds you well. I am writing to submit the revised version of our manuscript titled “Seasonal Fluctuations Impact on Intestinal Bacterial Communities and Pathogenic Diversity in Two Captive Dabbling Ducks” which was initially submitted to “Animals” under the manuscript ID animals-2722802. We sincerely thank you the reviewers for the insightful feedback and valuable suggestions provided during the review process.

We have carefully addressed each reviewer's comments and suggestions and all the changes are highlighted with yellow color in this revised version. In the following, we have attached a detailed response to the reviewers' comments, outlining how each comment was addressed, and would like to highlight the major changes we have made as follows:

Reviewer 2#

  1. Comment: Wording "seasonal fluctuations" - please be more specific- fluctuation of what? temperature, humidity...or use only word "season" in title

Response: We are very grateful for the reviewers’ constructive comments. We corrected it following your suggestion by use only “season” in the title. Please refer to the updated version for your review, Line 2.

Comment: It is necessary to correctly write the names of bacteria, genera, families, etc. throughout the text; write italics where necessary.

Response: We sincerely apologize for the oversight in not italicizing the families, genera, and species. Your feedback is highly valued. We have corrected them according to your suggestion. Please refer to the updated version for your review, Lines 243-250, 295-296, 314-317, 351.

  1. Comment: Determination of bacteria as "pathogenic" - why bacteria of the genus Escherichia-Shigella were not classified as pathogenic?

Response: We are sincerely thankful for your pointing question about the identification of pathogenic bacteria. We have decided to focus our analysis specifically on taxa at the species level.  Due to the incomplete species-level identification in the taxonomy results for the Escherichia-Shigella genus, we have chosen to exclude this genus from our analysis. This decision aims to ensure a precise assessment of the diversity and abundance of pathogenic bacteria species. Another reason for not identifying this genus as a pathogenic bacteria in this study is that members of the Escherichia genus are typically normal inhabitants of the intestinal tract in humans and other warm-blooded animals. They are largely benign, serving a nutritional role through the synthesis of vitamins [1]. For instance, Escherichia coli, a predominant nonpathogenic facultative flora in the human intestine, is generally harmless. It's important to note that only certain strains of E. coli have acquired the ability to cause disease [2].

References:

Beeckmans, S.; Xie, J.P. Glyoxylate Cycle☆. Reference Module in Biomedical Sciences. 2015. doi:10.1016/b978-0-12-801238-3.02440-5.

Guentzel, MN. Escherichia, Klebsiella, Enterobacter, Serratia, Citrobacter, and Proteus. In: Baron S, editor. Medical Microbiology. 4th edition. Galveston (TX): University of Texas Medical Branch at Galveston; 1996. Chapter 26. Available from: https://www.ncbi.nlm.nih.gov/books/NBK8035/.

  1. Comment: Title - please rephrase (possibly as "impact of season on..." and also please be more specific "two captive dabbling ducks" or two captive duck species"?

Response: Thank you for your valuable feedback. We have revised the title to align with your comment, and it is now: “Impact of Season on Intestinal Bacterial Communities and Pathogenic Diversity in Two Captive Duck Species”. Please refer to the updated version for your review, Lines 2-4.

  1. Comment: L 31-32 please rephrase this sentence (it is not clear "winter had sophisticated network structure"?)

Response: Thank you for your thoughtful comment. The sentence has already been rephrased. Please refer to the updated version for your review, Lines 32-33.

  1. Comment: L 37-38 please rephrase ("severe infection from the harmful diseases"?)

Response: Thank you for your thoughtful comment. The sentence has already been rephrased. Please refer to the updated version for your review, Lines 38-40.

  1. Comment: Chapter 2. Material and Methods - there is a clear need for more information regarding the birds and sampling (how many birds were used in this experiment (in L 91 it is stated "these two ducks")? how were they kept-inside, outside? what was their diet (only rice; L 320, 334?)

Response: We apologize for the unclear information and appreciate your comment. The experimented using a total of 12 ducks per species. Consequently, the total number of fecal samples analyzed for the study amounted to 96 (12 samples for each duck species and season). Those ducks were kept in closed cages which were divided into sleep areas and space areas for feeding. For a complete and balanced diet, we provided consistent commercial duck feed mixed with paddy rice for all seasons. Moreover, a water source for swimming grooming, and cooling down, as well as clean drinking water. We have corrected them. Please refer to the updated version for your review, lines 87-88, 96-97, and 337-338.

  1. Comment: L 89 what does it means that the samples were "suddenly put"?

Response: We thank you for your thoughtful comment. The term "suddenly put" refers to the immediate collection and placement of the stool sample into a plastic bag right after the ducks excrete it. Based on your feedback, we have made the necessary revisions. Please refer to the updated version for your review, Lines 90-94.

  1. Comment: L 106-108 please check, verb missing, reference?

Response: Thanks to the reviewer for pointing out the mistake. We have revised and added the reference for that sentence. Please refer to the updated version for your review, Lines 109-112.

  1. Comment: L 115 classifier?

Response: Thanks to the reviewer for pointing out the mistakes. We have revised it to an accurate term as per your suggestion. Please refer to the updated version for your review, Lines 119.

  1. Comment: L 133-134 please check - "that most differentially across"?

Response: Thank you for your thoughtful comment. The sentence has been revised. Please refer to the updated version for your review, Lines 139-140.

  1. Comment: L 162-165 please first write in full and than abbreviation in brackets

Response: We are very grateful for the reviewers’ constructive comments. We corrected it according to your suggestion. Please refer to the updated version for your review, Lines 167-170.

  1. Comment: L 198-200 please check - it is not clear "common teals exhibited a significant decrease"?

Response: We apologize for the unclear sentence and appreciate your comment. The sentence has been revised. Please refer to the updated version for your review, Lines 207-209.

  1. Comment: L 202 please check (it is only written "in Baikal teal"?)

Response: We apologize for the unclear sentence and appreciate your thoughtful comment. The sentence has been revised. Please refer to the updated version for your review, Lines 208-209.

  1. Comment: L 236 - please check (is the number of total of genera missing in this sentence?)

Response: Thanks to the reviewer for pointing out the mistake. The data has been added. Please refer to the updated version for your review, Lines 241.

  1. Comment: L 265-267 please check this sentence

Response: Thank you for your thoughtful comment. The sentence has been revised. Please refer to the updated version for your review, Lines 270-275.

  1. Comment: L 357 -please change "variation of seasonal variation"

Response: Appreciate the reviewer for pointing out the mistakes. The sentence has been revised. Please refer to the updated version for your review, Lines 362.

  1. Comment: In Discussion multiple sentences are not clear enough and should be rephrased (L 333-369, L 373-373, L 379-381, L 410-412, L 431-433, L 438-441, L 462-465)
    L 451 please change "is much higher level"

Response: We apologize for the unclear sentence and appreciate your comment. The sentences have been revised. Please refer to the updated version for your review.

L 333-369 >> Line 337-371

L 373-373 >> Line 374-375

L 379-381 >> Lines 380-382

L 410-412 >> Lines 412-414

L 431-433 >> Lines 435-437

 L 438-441 >> Lines 442-445

 L 462-465 >> Lines 466-469

L 451 >> Lines 455-456

We try our best to improve the manuscript and make some changes in the manuscript. And we have marked all the changes in yellow highlighted in the revised manuscript. We appreciate the Editors/Reviewers’ warm works earnestly and hope that the correction will meet with approval.

Once again, thank you very much for your comments and suggestions. I believe that these revisions have significantly enhanced the quality and contribution of our manuscript. We hope that these changes align well with the expectations of Animals and its readers. Please feel free to contact us if you need any further information or clarification. We are ready to provide any additional materials that may assist in the review process.

Thank you for your time and consideration.

Patthanan Sakda 1,2, Xingjia Xiang 1,2,3*, Zhongqiao Song 1, Yuannuo Wu1,2, and Lizhi Zhou 1,2,3*

1 School of Resources and Environmental Engineering, Anhui University, Hefei 230601, China

2Anhui Province Key Laboratory of Wetland Ecosystem Protection and Restoration, Anhui University, Hefei 230601, China

3Anhui Shengjin Lake Wetland Ecology National Long-Term Scientific Research Base, Chizhou 247230, China

* Correspondence: [email protected] (X.X.); [email protected] (L.Z.)

Reviewer 3 Report

Comments and Suggestions for Authors

Review of Seasonal Fluctuations Impact on Intestinal Bacterial Communities and Pathogenic Diversity in Two Captive Dabbling Ducks

This paper is readable and with some caveats the data support the conclusion.  I think it will make a good addition to the literature.  The methods are adequately described and appropriate.

The most significant thing needed from the authors is some information about the food consumed and the knowledge of the exclusivity of that consumption.   This must be added.

I have just a few thoughts from the paper which although minor do need to be addressed:

Abstract to state “under the instant  diet” does not communicate anything to the reader.  This needs to be modified –were the ducks fed the same food throughout the experiment?  This should be made clear in the abstract.  As they were wild the likely assumption is food is changing with the season.  Could this not be the reason for the changes recorded?

Introduction – line 61 needs to be reworded perhaps “much” or “little” if that is correct, should replace “have?”  On line 75-76 it states “consistent diet” and yet this seems unlikely – how does the diet not change with season?  Do you know the ducks consumed only the “steady regulated diet” stated on line 92?  Was it complete and balanced for ducks?  Please provide information on the diet.

Line 224 Baikal T\teal  seems to be a typographical error.

Comments on the Quality of English Language

some review needed for minor improvements.

Author Response

Response to reviewer 3# comments

Dear editor, associate editor, and reviewer

We hope this letter finds you well. I am writing to submit the revised version of our manuscript titled “Seasonal Fluctuations Impact on Intestinal Bacterial Communities and Pathogenic Diversity in Two Captive Dabbling Ducks” which was initially submitted to “Animals” under the manuscript ID animals-2722802. We sincerely thank you the reviewers for the insightful feedback and valuable suggestions provided during the review process.

We have carefully addressed each reviewer's comments and suggestions and all the changes are highlighted with yellow color in this revised version. In the following, we have attached a detailed response to the reviewers' comments, outlining how each comment was addressed, and would like to highlight the major changes we have made as follows:

Reviewer 3#

  1. Comment: Abstract to state “under the instant diet” does not communicate anything to the reader. This needs to be modified –were the ducks fed the same food throughout the experiment?  This should be made clear in the abstract.  As they were wild the likely assumption is food is changing with the season.  Could this not be the reason for the changes recorded?

Response: We are very grateful for your thoughtful comment. These ducks were consistently fed the same diet throughout the entire experiment. The abstract was revised for more clarity according to your suggestion. Please refer to the updated version for your review, Lines 25.

  1. Comment: line 61 needs to be reworded perhaps “much” or “little” if that is correct, should replace “have?”

Response: Thank you reviewer for pointing out the mistake. We already revised the sentence, please see lines 62-66 in the updated version.

  1. Comment: On line 75-76 it states “consistent diet” and yet this seems unlikely – how does the diet not change with season? Do you know the ducks consumed only the “steady regulated diet” stated on line 92?  Was it complete and balanced for ducks?  Please provide information on the diet.

Response: We apologize for the unclear information and appreciate your comment. For a complete and balanced diet, we provided consistent commercial duck feed mixed with paddy rice for all seasons. Moreover, we provided a water source for swimming grooming, and cooling down, as well as clean drinking water. We have corrected both the introduction and materials & methods sections, please see the lines 76-78, and 96-97 in the updated version.

  1. Comment: Line 224 Baikal T\teal seems to be a typographical error.

Response: We sincerely apologize for the typographical error and appreciate your valuable feedback. We have corrected it. Please refer to the updated version for your review, Line 230.

We try our best to improve the manuscript and make some changes in the manuscript. And we have marked all the changes in yellow highlighted in the revised manuscript. We appreciate the Editors/Reviewers’ warm works earnestly and hope that the correction will meet with approval.

Once again, thank you very much for your comments and suggestions. I believe that these revisions have significantly enhanced the quality and contribution of our manuscript. We hope that these changes align well with the expectations of Animals and its readers. Please feel free to contact us if you need any further information or clarification. We are ready to provide any additional materials that may assist in the review process.

Thank you for your time and consideration.

Patthanan Sakda 1,2, Xingjia Xiang 1,2,3*, Zhongqiao Song 1, Yuannuo Wu1,2, and Lizhi Zhou 1,2,3*

1 School of Resources and Environmental Engineering, Anhui University, Hefei 230601, China

2Anhui Province Key Laboratory of Wetland Ecosystem Protection and Restoration, Anhui University, Hefei 230601, China

3Anhui Shengjin Lake Wetland Ecology National Long-Term Scientific Research Base, Chizhou 247230, China

* Correspondence: [email protected] (X.X.); [email protected] (L.Z.)

Reviewer 4 Report

Comments and Suggestions for Authors

Title: Seasonal Fluctuations Impact on Intestinal Bacterial Communities and Pathogenic Diversity in Two Captive Dabbling Ducks

General Comments: The data presented from this study provide information for understanding the microbiome of captive wild ducks. The methods and analysis performed in the study were well-structured and scientific, and the authors presented sufficient literature that would corroborate their findings. However, the manuscript needs much improvement, particularly in presenting the information to its intended meaning.

Introduction

L46: delete 'an' in "...role in maintaining an animal's health..."

L61-64: please paraphrase or re-write the sentence to be less confusing and more direct to its intended meaning.

For example, do the authors mean, "In wild animals in the field, microbiome richness and diversity variations have been reported between different hosts. In particular, a significant composition of intestinal and pathogenic bacteria was found in sympatric Hooded crane (Grus monacha) and Greater White-Fronted Goose (Anser albifrons)." 

Materials & Methods

L88-90: Please paraphrase the sentence relating to the sampling collection and storage method. Also, do the authors mean "stored at" not "conserved at"?

L106-108. Please paraphrase or re-write this section to be clearer and easier to follow.

L111: Delete 'Then the' in "Then the quality filtering of ..."

L115: Change "classifier" to 'classified'

L119-120: Please include and mention the keywords used for the search

L134: The sentence "...most differentially ______ across sample groups." sounds incomplete. Please re-write the sentence to its intended meaning. 

Results

L164-165: Please correct the sentence to the intended meaning. WTC was mentioned twice. What about SPC?

L224: Please correct the words to their intended meaning, "...in Baikal T/ teal..."

L265-270: The paragraph is very lengthy and confusing. Please paraphrase and re-organize how the data are presented.

Example: The microbial co-occurrence pattern was explored using network analysis (Figure 6). Overall, the intestinal microbiota of Baikal teal (1,061 nodes; 70,271 edges) and common teal (1,252 nodes; 72,826 edges) had the most complex network during winter, followed by spring 

(B: 952 nodes, 41,618 edges; C: 961 nodes, 58,906 edges), summer () and autumn (). 

L288-313: Please italicize all necessary words presented as scientific names. 

DISCUSSION

L319-320: Regarding the line "under the same treatment and maintained on a diet," do the authors mean 'reared in the same condition' or please change to more appropriate words to describe the method.

L338-343: The sentence is long and confusing. Please re-write or paraphrase to its intended meaning to be more precise. 

L366-369: The sentence is long and confusing. Please re-write or paraphrase to its intended meaning to be more precise. 

L371: delete 'the' in "However, the LEfSe..."

L379-381: The sentence "Phylum Proteobacteria..." does not sound complete or is confusing. Please re-write or paraphrase to its intended meaning.

L410-419: Please re-write or paraphrase to its intended meaning to be more precise. 

L431-433: Please re-write to its intended meaning.

L462-465: Please re-write to its intended meaning.

L485-487: Please re-write the sentence.

Figure legends (4, 5,6): Please double-check that the proper legend was mentioned (i.e., WPC), and remove unnecessary legend or item not presented in the figure being described. 

Figure 5.: Please ensure the figure presents the data mentioned in the text—Figures 5 A, B, and C show legends for Baikal teal, but text mentions Fig A, B for Baikal and C, and D for common teal. 

Tables: Please ensure the titles and how they are mentioned in the text are appropriately labeled. 

Table S1. The identified potentially pathogenic bacteria carried by captive Baikal teal and common teal in this study: Please include related references. 

Comments on the Quality of English Language

Author Response

Response to reviewer 4# comments

Dear editor, associate editor, and reviewer

We hope this letter finds you well. I am writing to submit the revised version of our manuscript titled “Seasonal Fluctuations Impact on Intestinal Bacterial Communities and Pathogenic Diversity in Two Captive Dabbling Ducks” which was initially submitted to “Animals” under the manuscript ID animals-2722802. We sincerely thank you the reviewers for the insightful feedback and valuable suggestions provided during the review process.

We have carefully addressed each reviewer's comments and suggestions and all the changes are highlighted with yellow color in this revised version. In the following, we have attached a detailed response to the reviewers' comments, outlining how each comment was addressed, and would like to highlight the major changes we have made as follows:

Reviewer 4#

  1. Comment: L46: delete 'an' in "...role in maintaining an animal's health..."

Response: Thanks to the reviewer for pointing out the mistake. The word ‘an’ has been deleted based on your feedback. Please refer to the updated version for your review, Line 48.

  1. Comment: L61-64: please paraphrase or re-write the sentence to be less confusing and more direct to its intended meaning.

Response: Thank you for your thoughtful comment. The sentence has already been revised. Please refer to the updated version for your review, Lines 62-66.

  1. Comment: L88-90: Please paraphrase the sentence relating to the sampling collection and storage method. Also, do the authors mean "stored at" not "conserved at"?

Response: Thank you for your careful comment. The word ‘conserved at’ has been changed to ‘stored at’  and the sentence has been revised based on your feedback. Please refer to the updated version for your review, Line 93.

  1. Comment: L106-108. Please paraphrase or re-write this section to be clearer and easier to follow.

Response: We appreciate your thoughtful comment. The sentence has already been revised. Please refer to the updated version for your review, Lines 109-112.

  1. Comment: L111: Delete 'Then the' in "Then the quality filtering of ..."

Response: Thank you for your thoughtful comment. The word ‘Then the’ has been deleted based on your feedback. Please refer to the updated version for your review, Line 115.

  1. Comment: L115: Change "classifier" to 'classified'

Response: Thanks to the reviewer for pointing out the mistakes. We have revised it to an accurate term as per your suggestion. Please refer to the updated version for your review, Line 119.

  1. Comment: L119-120: Please include and mention the keywords used for the search

Response: Thank you for your valuable feedback. We have successfully mentioned the keywords and revised the sentence. Please refer to the updated version for your review, Lines 123-126.

  1. Comment: L134: The sentence "...most differentially ______ across sample groups." sounds incomplete. Please re-write the sentence to its intended meaning.

Response: We appreciate the thoughtful feedback. As per your comment, the sentence has already been revised. Please refer to the updated version for your review, Lines 139-140.

  1. Comment: L164-165: Please correct the sentence to the intended meaning. WTC was mentioned twice. What about SPC?

Response: We sincerely apologize for the oversight in the mistake of typo and sincerely appreciate your valuable feedback. We already changed WTC to SPC as per your comment. Please refer to the updated version for your review, Lines 167-170.

  1. Comment: L224: Please correct the words to their intended meaning, "...in Baikal T/ teal..."

Response: We sincerely apologize for the typographical error and appreciate your valuable feedback. We have corrected it. Please refer to the updated version for your review, Line 230.

  1. Comment: L265-270: The paragraph is very lengthy and confusing. Please paraphrase and re-organize how the data are presented. Example: The microbial co-occurrence pattern was explored using network analysis (Figure 6). Overall, the intestinal microbiota of Baikal teal (1,061 nodes; 70,271 edges) and common teal (1,252 nodes; 72,826 edges) had the most complex network during winter, followed by spring (B: 952 nodes, 41,618 edges; C: 961 nodes, 58,906 edges), summer () and autumn ().

Response: We greatly appreciate your valuable feedback. The sentence has been revised according to your suggestion. Please refer to the updated version for your review, Lines 270-275.

  1. Comment: L288-313: Please italicize all necessary words presented as scientific names.

Response: We sincerely apologize for the oversight in not italicizing the scientific name and appreciate your valuable feedback. The scientific names including genera and families have been italicized. Please refer to the updated version for your review, Lines 243-250, 295-296, 314-317, 351.

  1. Comment: L319-320: Regarding the line"under the same treatment and maintained on a diet," do the authors mean 'reared in the same condition' or please change to more appropriate words to describe the method.

Response: We express gratitude for your insightful commentary. Indeed, we have aligned our revisions with your feedback that the ducks were reared in the same condition. We already revised it as per your suggestion. Please refer to the updated version for your review, Lines 324.

  1. Comment: L338-343: The sentence is long and confusing. Please re-write or paraphrase to its intended meaning to be more precise. 

Response: Thank you for your thoughtful comment. The sentence has been revised. Please refer to the updated version for your review, Lines 343-349.

  1. Comment: L366-369: The sentence is long and confusing. Please re-write or paraphrase to its intended meaning to be more precise. 

Response: We appreciate your considerate feedback. The sentence has already been revised. Please refer to the updated version for your review, Lines 367-371.

  1. Comment: L371: delete 'the' in "However, the.."

Response: Thank you for pointing out the mistake. The word ‘the’ in the sentence has already been deleted. Please refer to the updated version for your review, Line 373.

  1. Comment: L379-381: The sentence "Phylum Proteobacteria..." does not sound complete or is confusing. Please re-write or paraphrase to its intended meaning.

Response: Appreciation for your insightful comment. The sentence has already been revised. Please refer to the updated version for your review, Lines 380-382.

  1. Comment: L410-419: Please re-write or paraphrase to its intended meaning to be more precise. 

Response: We appreciate your thoughtful comment. The sentence has already been revised. Please refer to the updated version for your review, Lines 412-423.

  1. Comment: L431-433: Please re-write to its intended meaning.

Response: Thank you for your thoughtful comment. The sentence has already been revised. Please refer to the updated version for your review, Lines 435-437.

  1. Comment: L462-465: Please re-write to its intended meaning.

Response: Thank you for your thoughtful comment. The sentence has already been revised. Please refer to the updated version for your review, Lines 466-469.

  1. Comment: L485-487: Please re-write the sentence.

Response: Thank you for your thoughtful comment. The sentence has already been revised. Please refer to the updated version for your review, Lines 492-494.

  1. Comment: Figure legends (4, 5,6): Please double-check that the proper legend was mentioned (i.e., WPC), and remove unnecessary legend or item not presented in the figure being described. 

Response: We sincerely apologize for the typographical error and appreciate your valuable feedback. The legends of Figure 4,5,6 have been revised, by removing the items not presented from the Figure 4 legend, and changing the double WTC (winter common teal) to SPC (spring common teal). Please refer to the updated version for your review, Lines 238-239, 267-268, and 283.

  1. Comment: Figure 5.: Please ensure the figure presents the data mentioned in the text—Figures 5 A, B, and C show legends for Baikal teal, but text mentions Fig A, B for Baikal and C, and D for common teal. 

Response: We sincerely apologize for the mistake and appreciate you pointing it out. The incorrect legends were mentioned (ATB, SMB, SPC, WTC) in Figure 5C, which presents common teal NMDS. The figure has been revised to correct legends (ATC, SMC, SPC, WTC) and corresponds with the data. Please refer to Figure 5C in the updated version for your review.

  1. Comment: Tables: Please ensure the titles and how they are mentioned in the text are appropriately labeled. 

Response: Thank you for your informative suggestion. Table S3 had been revised for the more appropriate title and the label of Table S8 was revised to corrected data (changed double WTC to SPC). Please refer to Table S3 and Table S8 in the Supplementary Material file updated version for your review.

  1. Comment: Table S1. The identified potentially pathogenic bacteria carried by captive Baikal teal and common teal in this study: Please include related references. 

Response: We greatly appreciate your valuable feedback. The references for identifying pathogenic bacteria have been added in Table S1 and the references section [93-99] following your suggestion. Please refer to the updated version for your review.

We try our best to improve the manuscript and make some changes in the manuscript. And we have marked all the changes in yellow highlighted in the revised manuscript. We appreciate the Editors/Reviewers’ warm works earnestly and hope that the correction will meet with approval.

Once again, thank you very much for your comments and suggestions. I believe that these revisions have significantly enhanced the quality and contribution of our manuscript. We hope that these changes align well with the expectations of Animals and its readers. Please feel free to contact us if you need any further information or clarification. We are ready to provide any additional materials that may assist in the review process.

Thank you for your time and consideration.

Patthanan Sakda 1,2, Xingjia Xiang 1,2,3*, Zhongqiao Song 1, Yuannuo Wu1,2, and Lizhi Zhou 1,2,3*

1 School of Resources and Environmental Engineering, Anhui University, Hefei 230601, China

2Anhui Province Key Laboratory of Wetland Ecosystem Protection and Restoration, Anhui University, Hefei 230601, China

3Anhui Shengjin Lake Wetland Ecology National Long-Term Scientific Research Base, Chizhou 247230, China

* Correspondence: [email protected] (X.X.); [email protected] (L.Z.)

Round 2

Reviewer 2 Report

Comments and Suggestions for Authors

Dear authors,

thank you for taking into consideration all the comments given by the reviewers.

Best regards,

Comments on the Quality of English Language

Minor changes needed